# Pneumatosis Intestinalis Induced by Anticancer Treatment: A Systematic Review

**DOI:** 10.3390/cancers14071666

**Published:** 2022-03-25

**Authors:** Gianluca Gazzaniga, Federica Villa, Federica Tosi, Elio Gregory Pizzutilo, Stefano Colla, Stefano D’Onghia, Giusy Di Sanza, Giulia Fornasier, Michele Gringeri, Maria Victoria Lucatelli, Giulia Mosini, Arianna Pani, Salvatore Siena, Francesco Scaglione, Andrea Sartore-Bianchi

**Affiliations:** 1Department of Medical Biotechnology and Translational Medicine, Postgraduate School of Clinical Pharmacology and Toxicology, Università degli Studi di Milano, 20122 Milan, Italy; gianluca.gazzaniga@unimi.it (G.G.); stefano.colla@unimi.it (S.C.); stefano.donghia@unimi.it (S.D.); giusy.disanza@unimi.it (G.D.S.); giulia.fornasier@unimi.it (G.F.); michele.gringeri@unimi.it (M.G.); mariavictoria.lucatelli@unimi.it (M.V.L.); giulia.mosini@unimi.it (G.M.); 2Department of Oncology and Hemato-Oncology, Università degli Studi di Milano, 20122 Milan, Italy; federica.villa@ospedaleniguarda.it (F.V.); elio.pizzutilo@unimi.it (E.G.P.); arianna.pani@ospedaleniguarda.it (A.P.); salvatore.siena@unimi.it (S.S.); francesco.scaglione@unimi.it (F.S.); 3Niguarda Cancer Center, Grande Ospedale Metropolitano Niguarda, 20162 Milan, Italy; federica.tosi@ospedaleniguarda.it; 4Department of Chemical-Clinical and Microbiological Analyses, Grande Ospedale Metropolitano Niguarda, 20162 Milan, Italy

**Keywords:** pneumatosis intestinalis, oncology, hematology, chemotherapy, molecular targeted therapy, immunotherapy, adverse drug events

## Abstract

**Simple Summary:**

Anticancer treatments commonly cause adverse events (AE). Among others, pneumatosis intestinalis (PI) is reported to be infrequent, though it can lead to severe consequences. The aim of our systematic review was to investigate the concurrency of PI and oncological therapy exposure; moreover, we defined the characteristics of patients and the primarily involved tumor types. We analyzed 88 different episodes of PI. The median time of onset was 6 weeks and oncological patients with respiratory system cancers and those treated with targeted therapies appeared be at higher risk. Symptoms were frequently mild to absent; nevertheless, life-threatening complications were reported. Therefore, this AE, although uncommon, should be considered in the case of specific symptoms. Potential pharmacological mechanisms of anticancer drugs in inducing PI are also discussed.

**Abstract:**

Pneumatosis intestinalis (PI) is a rare condition due to the presence of gas within the bowel wall; it is mainly caused by endoscopic procedures, infections and other gastrointestinal diseases. Oncological therapies have been reported to be a cause of PI as well, but their role is not clearly defined. This systematic review investigates the concurrency of PI and antitumor therapy in cancer patients, considering both solid tumors and onco-hematological ones. We performed a literature review of PubMed, Embase and the Web of Science up to September 2021 according to the PRISMA guidelines. A total of 62 papers reporting 88 different episodes were included. PI was mainly reported with targeted therapies (sunitinib and bevacizumab above all) within the first 12 weeks of treatment. This adverse event mostly occurred in the metastatic setting, but in 10 cases, it also occurred also in the neoadjuvant and adjuvant setting. PI was mostly localized in the large intestine, being fatal in 11 cases, while in the remaining cases, symptoms were usually mild, or even absent. A significant risk of PI reoccurrence after drug reintroduction was also reported (6/18 patients), with no fatal outcomes. Potential pharmacological mechanisms underlying PI pathogenesis are also discussed. In conclusion, although uncommonly, PI can occur during oncological therapies and may lead to life-threatening complications; therefore, consideration of its occurrence among other adverse events is warranted in the presence of clinical suspicion.

## 1. Introduction

### 1.1. Pneumatosis Intestinalis

Pneumatosis intestinalis (PI), also known as pneumatosis cystoides intestinalis, intramural gas, pneumatosis coli, pseudolipomatosis, bullous emphysema of the intestine and lymphopneumatosis, is a condition caused by the presence of gas within the bowel wall, specifically in three layers (mucosa, submucosa and subserosa), and it can also be associated with pneumoperitoneum [1,2].

The incidence of PI is difficult to estimate, since most patients are asymptomatic, but the overall incidence based on autopsy series has been reported to be 0.03% [3].

The pathogenesis of PI is not fully understood yet, as it is probably multifactorial and many hypotheses have been proposed; i.e., a mechanical theory speculates that the air penetrates into the bowel wall through breaks in the luminal surface or through mesenteric blood vessels, whereas a bacterial theory blames gas-forming bacteria penetrating the submucosa as the agents responsible for PI’s occurrence [4,5]. Finally, a biochemical theory has been formulated, according to which the excess of hydrogen gas due to the fermentation of food by luminal bacteria produces an increase in the intestinal lumen pressure, thus forcing gas directly through the mucosa [6]. Nevertheless, some medications have been associated with the risk of PI, even though the pharmacological mechanisms have not been elucidated yet [7].

PI can be idiopathic (15%) or secondary (85%) [2]. Among adult patients, secondary PI is mainly associated with gastrointestinal disease, pulmonary diseases, mechanical ventilation, endoscopic procedures, infections and other immunological conditions [8]. Medications used for cancer treatment are among the most recognized causes of PI, including cytotoxic chemotherapeutic agents [9] and molecular targeted agents [10,11,12,13,14,15,16,17,18,19,20,21,22,23,24,25,26,27,28,29,30,31,32,33,34,35,36,37,38,39,40,41,42,43,44,45,46,47,48,49,50,51,52,53,54,55].

Though PI is often a benign incidental finding, in some patients it may determine abdominal pain, obstruction, bleeding, vomiting, weight loss, diarrhea or constipation, flatulence and loss of appetite, up to life-threatening conditions, usually due to intraabdominal complications, such as bowel obstruction, perforation, volvulus, intussusception, hematochezia due to ulceration of the mucosa, necrotic bowel and pneumoperitoneum, which occur in approximately 3% of total cases [56,57].

PI can only be detected by imaging techniques, such as abdominal radiography and contrast-enhanced abdominal computed tomography (CT) or by endoscopy, which allows for finding cysts or air into the bowel wall. Radiological findings of PI consist in the presence of low-density linear or bubbly pattern of gas in the bowel wall. CT scans have been shown to be the most sensitive technique, especially for detecting features suggestive for more advanced PI stages, such as hepatic portal and portomesenteric venous gas [3] (Figure 1).

Laboratory tests are of limited value in diagnosing PI since they are usually normal [58].

### 1.2. Management of PI

Patients with signs of peritonitis, metabolic acidosis, lactate >2.0 mmol/L or portal venous gas should be subjected to an emergent exploratory laparotomy [8]. Other patients do not require emergent management and the treatment will depend on the severity of symptoms; asymptomatic patients usually do not require any treatment, whereas patients with mild symptoms are generally managed with antibiotics and an elemental diet, and patients with moderate/severe symptoms may require hospitalization, antibiotics, diet and oxygen therapy. In some cases, hyperbaric oxygen therapy could be necessary.

Moreover, patients who remain symptomatic despite medical therapy or patients who develop complications, such as bowel obstruction or peritonitis, should undergo surgery. Endoscopic therapy may be a valid option for those patients who cannot be operated on because of high perioperative risk.

Clearly, the underlying cause must be found, and patients should receive therapy directed against the etiology of PI; if a pharmacological therapy is held responsible for the PI occurrence, it must be promptly discontinued [6,59].

Antibiotic treatment should be stopped only after clinical and radiological resolution of PI, and the most commonly used one is metronidazole. Those patients for whom metronidazole is not an available option, ampicillin, tetracycline or vancomycin can be used [60,61].

Although several approaches have been implemented in order to treat patients with symptomatic PI, the evidence to support their efficacy is still limited to case reports or at best small case series, which do not directly compare these interventions.

### 1.3. Pneumatosis Intestinalis and Oncological Therapies

Bowel toxicity is usually associated with standard cytotoxic chemotherapy or targeted therapy (TT), involving either monoclonal antibodies (mAb) or kinase inhibitors (KIs). Despite this, there have only been a few rare cases where cancer patients have been reported to develop PI and/or bowel perforation as a complication of chemotherapy (including cyclophosphamide, cytarabine, vincristine, doxorubicin, daunorubicin, etoposide, docetaxel, irinotecan and cisplatin) or TT, including bevacizumab, sunitinib and sorafenib [7,37,62]. Immunotherapy is known to be a relevant cause of gastrointestinal toxicities as well, but PI cases are anecdotic [63,64,65].

### 1.4. Aim of the Study

The aim of the study was to investigate the concurrency of PI and oncological treatments in cancer patients.

## 2. Materials and Methods

### 2.1. Search Strategy

The systematic review was conducted in accordance with the Preferred Reporting Items for Systematic Reviews and Meta-Analyses (PRISMA) guidelines for the evidence of pneumatosis intestinalis following antitumoral pharmacological therapy as reported in the scientific literature [66].

A literature search of PubMed/MEDLINE, Embase and Web of Science was performed up to 30 September 2021. We searched articles using the following main keywords: “pneumatosis intestinalis”, “molecular targeted therapy”, “immunotherapy” and “chemotherapy”. We combined terms with the Boolean operators “AND” and “OR”.

Search strategy was adapted as necessary for each database, and complete details of each search are described in Appendix A.

This review has been registered in PROSPERO register, ID number 316286.

The whole search process was performed by two independent groups of researchers: a pharmacologist team and an oncologist one, providing a multifaceted context of expertise. Disagreements about eligibility were resolved through discussion with the senior author.

### 2.2. Eligibility Criteria

Inclusion criteria were the following: clinical trials, observational studies, case series or case reports, written in English, Spanish or French, that reported cases of oncological or hematological patients in which PI occurred while they were undergoing antitumoral pharmacological treatments.

Studies were excluded if they: (1) reported on pediatric patients, (2) were referred to patients who had previously undergone allogeneic or autologous stem cells transplantation to exclude graft-versus-host-disease-induced PI.

### 2.3. Outcomes

For each PI occurrence in a cancer patient undergoing oncology treatment, we collected data regarding clinical characteristics, tumor histology, type of cancer therapy, time-to-onset of PI and clinical outcome.

### 2.4. Study Selection and Data Extraction

We imported all the retrieved references into EndNote, and duplicates were removed. Titles and abstracts of potentially eligible articles were screened. Full-text versions of the selected articles were obtained and assessed for eligibility based on our prespecified eligibility criteria.

The authors independently extracted and tabulated the following data from each case report or case series: patient characteristics (age, sex, ethnicity and tumor disease details, comorbidities), therapy information (treatment and concomitant drugs) and information related to the AE (time to onset, site of pneumatosis intestinalis, potential intervention and outcome). We identified additional relevant articles through the examination of references cited in the case series or case report reviewed.

## 3. Results

Our search resulted in 236 unique titles. All titles were screened, after which 65 full-text articles were assessed for eligibility. Ultimately, 50 records that met our eligibility criteria were identified, and 12 more reports were retrieved through reference mining, reaching a total of 62 articles that were included in our systematic review (Figure 2) [9,10,11,12,13,14,15,16,17,18,19,20,21,22,23,24,25,26,27,28,29,30,31,32,33,34,35,36,37,38,39,40,41,42,43,44,45,46,47,48,49,50,51,52,53,54,55,62,64,65,67,68,69,70,71,72,73,74,75,76,77,78]. A full list of references of the included articles is available in Appendix A. Neither clinical trials nor observational studies were found in our search.

Table 1 summarizes the characteristics of all 88 patients described in case reports and case series included in the systematic review: 72 were oncological patients and 16 hematological. The mean age of patients was 65.3 ± 10.5 and 49.6 ± 17.5 years, for oncological and hematological patients, respectively. Females represented 45.8% of oncological patients and 37.5% of hematological patients. Comorbidities were reported in only 18 cases (20.5%) in 15 oncological and 3 hematological patients. Further information concerning comorbidities is available in Appendix A.

Regarding oncological patients, the tumor site involved the respiratory system (29.2%), gastrointestinal (GI) system (27.8%), the urinary system (15.3%), the reproductive system (5.6%), head and neck (6.9%), breast (5.6%), the endocrine system (5.6%), the nervous system (2.8%) or the skin (1.4%); in 69.4% of cases, PI occurred in stage IV cancer patients. Concerning hematological patients, 31.3% were affected by non-Hodgkin lymphoma, 25.0% by acute myeloid leukemia or by acute lymphoblastic leukemia, 12.5% by chronic myeloid leukemia and 6.3% by Hodgkin lymphoma.

Pharmacological treatment used in the clinical cases found in our search included standard cytotoxic therapy, specifically in 27.8% of oncological cases and in 75.0% of hematological cases, and TT, including mAbs and KIs in 52.8% of oncological and 18.8%, respectively, of hematological cases. Some patients underwent association treatment, TT and cytotoxic therapy (two oncological patients and one hematological patient), radiotherapy and cytotoxic agents (five oncological patients) or radiotherapy and TT (five oncological patients). PI was mainly reported with the use of targeted therapies (sunitinib and bevacizumab above all) rather than cytotoxic treatment. Among oncological patients, in the vast majority, PI was mainly observed during treatment in the metastatic setting (69.4%) rather than in the neoadjuvant (11.1%) or adjuvant (2.8%) context. Indeed, in the metastatic setting, the use of targeted agents was also more frequent (68.0% vs. 0%, *p* = 0.0001).

The drugs most involved in PI occurrence in these case reports were fluorouracil (9 patients) and cyclophosphamide (8 patients) as for cytotoxic agents, and sunitinib (10 patients), cetuximab (9 patients) and bevacizumab (8 patients) for TT (Figure 3).

Details about pharmacological treatment are reported in Table 2. Only six patients underwent GI tract surgery in the days before PI onset.

The median time to PI onset was 6 weeks (data were available for 53 cases), and 41 out of 53 patients developed PI within 12 weeks of starting treatment (Figure 4).

Concerning oncological patients, 29.2% were asymptomatic, whereas only 12.5% of hematological patients did not manifest symptoms. For asymptomatic patients, PI was discovered during routine follow-up, or it was an incidental finding on a CT scan performed for other clinical reasons. Perforation occurred in 11 cases: 10 oncological patients (3 lung cancer, 3 renal carcinoma, 1 breast cancer, 1 colorectal cancer, 1 esophageal cancer and 1 SCC) and 1 hematological and was fatal in 5 of these cases. However, PI was proved to be fatal in 11 cases overall, 9 of which were oncological and 2 of which were hematological patients; one of them was undergoing neoadjuvant chemoradiotherapy.

PI was most localized in the large intestine (44.3%), followed by the small intestine (26.1%), while it involved the entire GI tract in 21.6% cases.

PI was mostly treated with antimicrobial therapy (metronidazole), parenteral nutrition and oxygen support (Figure 5). After PI resolution, in 18.1% and 31.3% of oncological and hematological cases, respectively, therapy was restarted, but symptoms occurred again after reintroduction in 6/18 patients (33.3%). More details about PI manifestation and management are reported in Table 3 and Figure 5.

## 4. Discussion

PI is a rare yet clinically relevant condition that, even when asymptomatic at diagnosis, can lead to life-threatening complications. The presence of gas within the bowel wall could be caused by mechanical conditions (e.g., obstruction, increased abdominal pressure caused by surgery), infections and most commonly by idiopathic causes. Importantly, in cancer patients PI can be induced by treatment, and the underlying etiology has not been fully elucidated yet. To our knowledge, this is the first systematic review of cases of PI occurring during antitumor therapy in patients with solid and hematologic malignancies.

From our analysis, this complication more frequently occurs after TT exposure, including monoclonal antibodies (mAbs) and kinase inhibitors (KIs) (51.1% overall), being higher in patients with solid rather than hematologic tumors. On the other hand, we found PI occurring after cytotoxic therapy in 36.4% of patients overall, this being more commonly in hematological patients. Only two patients reported PI following immunotherapy, alone or in combination with TT (2.8%). All other cases (14.8%) were patients undergoing a combination of chemotherapy, TT and/or radiotherapy.

The underlying pharmacologic mechanisms are not fully understood.

The intestine is the most highly regenerative organs in the human body, regenerating its epithelium every five to seven days. Under normal conditions, epithelial proliferation, differentiation and apoptosis are tightly regulated and are therefore finely balanced. Its homeostasis can be interfered with by several elements, such as cytostatic drugs, leading to alterations in epithelial structure and cell functions, resulting in mucosal damage. Indeed, cytotoxic agents act on proliferating cells, inhibiting specific cellular processes according to their mechanisms of action, without any distinction between tumoral and nontumoral cells. Therefore, bowel epithelium is often impaired, given the high frequency of cell turnover [29,30]. Moreover, the mechanisms behind PI occurrence during cytotoxic treatment could reside in the influence it exerts on tissue repair. Under physiological conditions, the epithelial tissue damage triggers an inflammatory process that attracts several cells from the bone marrow, such as granulocytes, monocytes and lymphocytes. This process triggers specific signal pathways that are directly implicated in the regenerative process of the damaged tissue; since myelosuppressive drugs induce bone marrow aplasia, this results in the loss of hematopoietic cells, thus hindering the regeneration process of damaged tissues.

Conversely, TTs react with specific molecular targets and are expected to cause fewer side effects [79]. However, PI was recently described in patients receiving TT, such as anti-VEGF/VEGFR, anti-EGFR, anti-PDGFR and c-KIT inhibitors. Since VEGF has a key role in angiogenesis, the inhibition of this substrate or its receptor may lead to the decreased capillary density of the intestine, injuring the efficient gas diffusion across the intestinal wall and blood vessels. Furthermore, reduced vascularization leads to the development of ischemia. These conditions may lead to PI occurrence and in most serious cases may enhance bowel perforation. As for anti-EGFR drugs, EGF is crucial for epithelial cells’ regeneration; deficient signaling of the EGFR may cause a lower efficiency in damage bowel repair. PDGF is also involved in tissue reparation, promoting clot generation; hence, the block of the signal may be involved in PI’s occurrence.

Furthermore, c-KIT is the receptor of the stem cell factor (SCF) responsible for proliferation and differentiation of hematopoietic cells. PI may be promoted by the inhibition of the reparation process governed by the inflammatory system. In addition, c-KIT receptors are located on interstitial cells of Cajal (ICC), which are involved in the generation of spontaneous electrical slow waves in the GI tract. Reduced bowel motility may facilitate PI onset [55].

Despite the widespread use of immunotherapy over the last decade [80], we found only one case of PI in a patient following a monotherapy with an immune checkpoint inhibitor (PD-L1 inhibitor). In another case, PI occurred in a patient concurrently treated with a PD-1 blocker and a multikinase inhibitor. Notably, PD-1 is expressed on T cells upon activation, while PD-L1 is physiologically expressed in different tissues, but it has a main role only in lymphocytes and dendritic cells. No significant PD-L1 expression is known to occur on gastrointestinal cells; therefore, a direct role of immunotherapy in PI causation is hard to explain [81]. Moreover, monoclonal antibodies have high selectivity for their targets; hence, binding collateral receptors is unlikely. [82,83] In addition, an indirect action may be hypothesized; the gastrointestinal wall is naturally rich in immune cells that guarantee protection against luminal pathogens. Immunotherapy is known to cause general immune system activation, not only in tumoral masses but in blood vessels too [81]. That may lead to vessels’ alteration and air penetration through bowel wall breaks.

Potentially, all these mechanisms might even have a cumulative effect and interact, reciprocally provoking PI.

We know from the literature that there are risk factors and predisposing diseases associated with PI, such as diabetes and alpha-glucosidase inhibitor treatment [84], COPD and asthma [85], diseases affecting GI motilities and mucosal integrity such as inflammatory bowel disease [86]. We can assume that tumor involvement of the gastrointestinal tract could be a risk factor for PI due to the tumor alteration of the intestinal wall integrity. Indeed, in our review 22.7% of both oncological and hematological cases had gastrointestinal tract tumor involvement.

Among oncological patients, we observed that PI can occur both during adjuvant treatment (eight patients, 11.1%) and neoadjuvant treatment (two patients, 2.8%). Those percentages are relevant if we consider the curative treatment setting and even more so if we take into account the risk of serious and even fatal outcomes of this AE (12.5% of patients).

Another important and clinically relevant aspect is drug reintroduction after PI. In 18 cases, after PI resolution, therapy was restarted. PI relapse occurred in 6/18 patients (33.3%), with no fatal outcomes. Among relapsed patients, five were oncological patients and the drug involved was gefitinib in two cases, erlotinib and sorafenib in one case and cytotoxic chemotherapy in one case; the sixth relapse occurred in a hematological patient exposed to a combination of chemo and TT (imatinib). Considering the treatment setting and the potential expected clinical benefit from the drug reintroduction, we suggest a careful evaluation of restarting treatment, paying attention to early symptoms, with a strict radiological follow-up.

Strengths of our study include the comprehensive and systematic search and that it was performed by two independent groups with different and complementary backgrounds: oncologists and pharmacologists.

Nevertheless, our study has several limitations. First, because of the low incidence of PI, our sources were mainly case reports and case series, and therefore we cannot draw conclusions about the actual incidence or prevalence of PI due to cancer treatment in the general patient population, nor we can establish the inference of causality and risk. In addition, often case reports included in our study displayed an overall poor quality, missing several important pieces of information; for instance, as shown in Table 1, only one-fifth of cases reported comorbidities. These data were indeed available in 18/88 cases, and among those, eight patients had concomitant diseases (diabetes, asthma, COPD and interstitial lung disease) or had undergone procedures (mechanical ventilation) that could be potential primary causes or, at least, relevant risk factors of PI [85]. Finally, in addition, information regarding other non-oncological therapies taken by patients was often missing; therefore, a drug–drug interaction analysis was not performed. These conditions led to a mere description of cases, and neither a correlation nor causation assessment could be established.

## 5. Conclusions

PI is an uncommon, yet clinically relevant condition described in cancer patients undergoing anticancer pharmacological treatments that, even when asymptomatic, can lead to life-threatening complications. This condition can take place mainly following TT and cytotoxic, with potential cumulative risk with the presence of known predisposing factors. Our analysis reveals a rapid time of onset; therefore, higher caution is suggested within the first 12 weeks.

## Figures and Tables

**Figure 1 cancers-14-01666-f001:**
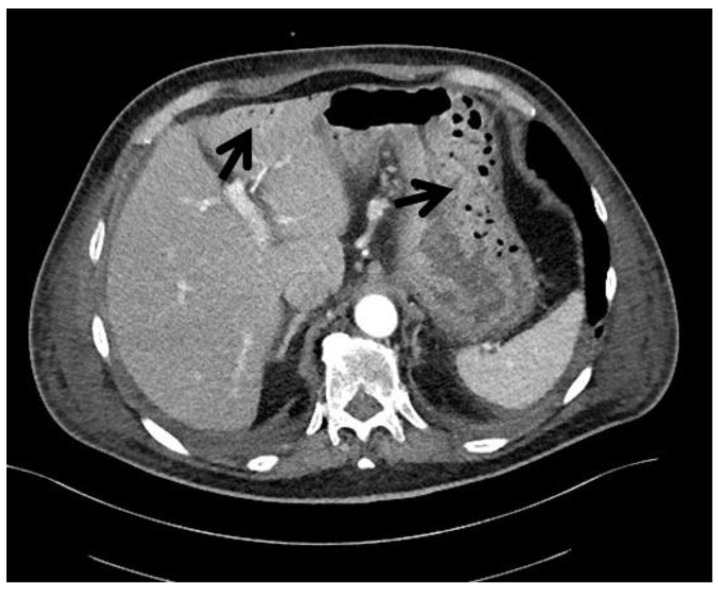
Radiological findings in a 79-year-old patient diagnosed at our institution with a gastrointestinal stromal tumor (GIST) and symptoms of abdominal pain. CT scan shows the presence of gas in the gastric wall at the greater curvature and in left intrahepatic portal system (black arrows). (Courtesy of Prof. Angelo Vanzulli, Radiology Department, Grande Ospedale Metropolitano Niguarda, Milano, Italy).

**Figure 2 cancers-14-01666-f002:**
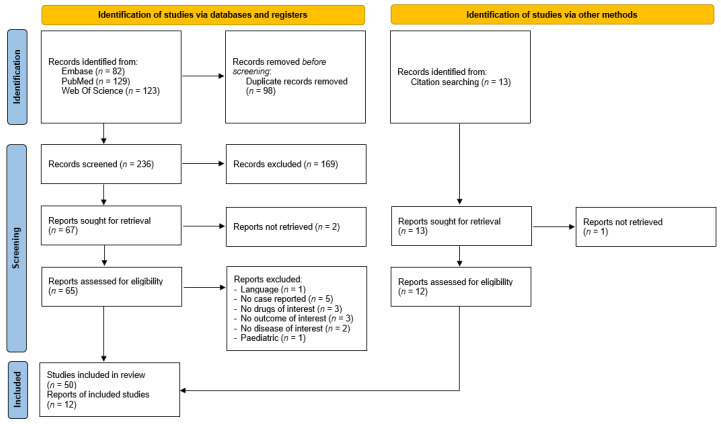
PRISMA Flow.

**Figure 3 cancers-14-01666-f003:**
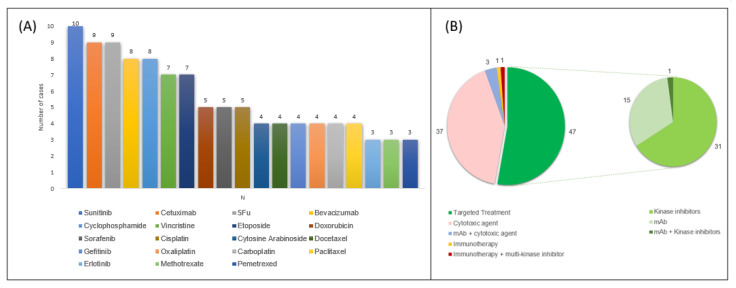
Anticancer therapies most commonly reported in published cases of cancer patients with PI (Panel (**A**), only reported if occurrence was found in at least 3 patients) and number of cases grouped according to pharmacological class (Panel (**B**)) 5FU, fluorouracil; mAb monoclonal antibodies.

**Figure 4 cancers-14-01666-f004:**
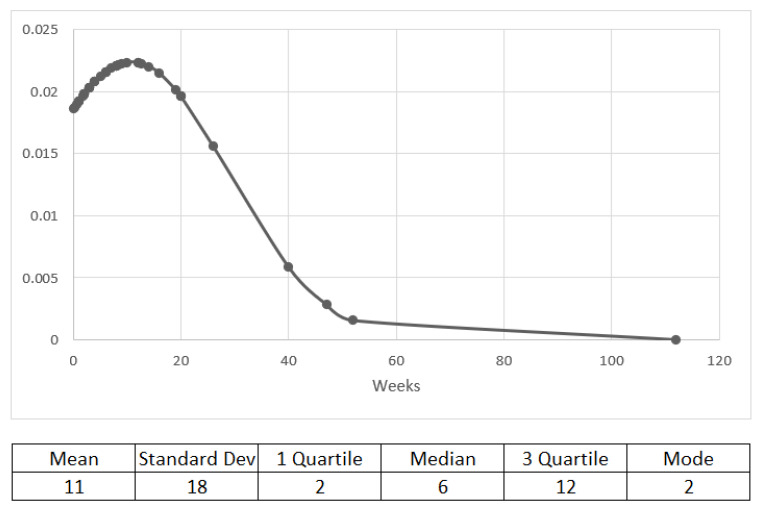
Time (in weeks) from treatment start to PI onset.

**Figure 5 cancers-14-01666-f005:**
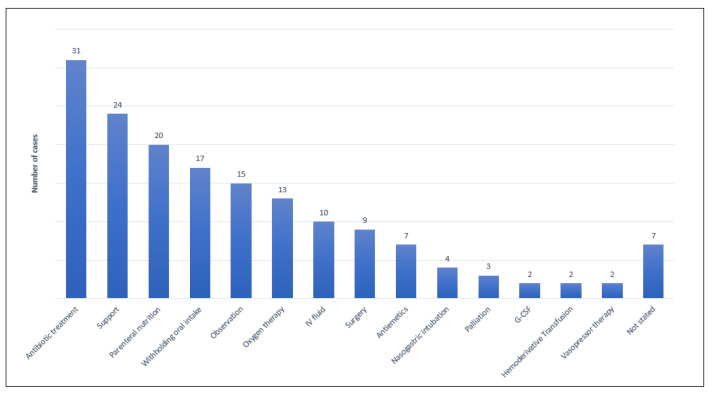
Overview of pneumatosis intestinalis management.

**Table 1 cancers-14-01666-t001:** Patients’ characteristics.

Patients Characteristics	Oncological (72 Cases)	Hematological (16 Cases)	Total (88 Cases)
Age (mean ± SD)—yr	65.3 ± 10.5	49.6 ± 17.5	62.4 ± 13.5
Females—no (%)	33 (45.8)	6 (37.5)	39 (44.3)
**Primary tumor site/classification—no (%)**			
**Endocrine system**			
pNET	2 (2.8)	-	
Papillary thyroid carcinoma	2 (2.8)	-	
**Gastrointestinal system**			
Cholangiocarcinoma	1 (1.4)	-	
Colorectal cancer	7 (9.7)	-	
Esophagus adenocarcinoma	5 (6.9)	-	
GIST	3 (4.2)	-	
Hepatocellular carcinoma	2 (2.8)	-	
PDAC	2 (2.8)	-	
**Head and neck**			
HNSCC	5 (6.9)	-	
**Blood and lymphatic system**			
ALL	-	4 (25.0)	
AML	-	4 (25.0)	
CML	-	2 (12.5)	
Hodgkin lymphoma	-	1 (6.3)	
Non-Hodgkin lymphoma	-	5 (31.3)	
**Nervous system**			
Cerebellar subependymoma	1 (1.4)	-	
GBM	1 (1.4)	-	
**Urinary system**			
Bladder carcinoma	1 (1.4)	-	
Renal cell carcinoma	10 (13.8)	-	
**Reproductive system**			
Uterine cancer	2 (2.8)	-	
Ovarian cancer	1 (1.4)	-	
Testis neoplasm	1 (1.4)	-	
**Breast**			
Breast cancer	4 (5.6)	-	
**Respiratory system**			
Lung adenocarcinoma	20 (27.8)	-	
Pleural mesothelioma	1 (1.4)	-	
**Skin**			
SCC	1 (1.4)	-	
**Settings—no (%)**			
Metastatic setting—no (%)	50 (69.4)	-	
Adjuvant setting—no (%)	8 (11.1)	-	
Neoadjuvant setting—no (%)	2 (2.8)	-	
Unknown setting—no (%)	12 (16.7)	-	
**Pharmacological therapy—no (%)**			
Cytotoxic agents	20 (27.8)	12 (75.0)	32 (36.4)
Targeted therapy (mAb + KIs)	38 (52.8)	3 (18.8)	41 (46.6)
Immunotherapy	1(1.4)	-	1 (1.1)
Immunotherapy + targeted therapy	1 (1.4)	-	1 (1.1)
Targeted Therapy + cytotoxic	2 (2.8)	1 (6.3)	3 (3.4)
Radiotherapy + cytotoxic	5 (6.9)	-	5 (5.7)
Radiotherapy + targeted therapy	5 (6.9)	-	5 (5.7)
**Reported comorbidity—no (%)**	15 (20.8)	3 (18.8)	18 (20.5)

ALL: acute lymphoblastic leukemia; AML: acute myeloid leukemia; CML: chronic myeloid leukemia; COPD: chronic obstructive pulmonary disease; GBM: glioblastoma multiforme; GIST: gastrointestinal stromal tumors; HNSCC: head and neck squamous cell carcinoma; KIs: kinase inhibitors; mAb: monoclonal antibodies; pNET: pancreatic neuroendocrine tumors; PDAC: pancreatic ductal adenocarcinoma; SCC: squamous cell carcinoma; SD: standard deviation.

**Table 2 cancers-14-01666-t002:** Pharmacological therapy suspected of causing PI.

Pharmacological Therapy—No (%)		Oncological—No (%)	Hematological—No (%)
**Cytotoxic agents**		25 (34.7)	12 (75.0)
**mAb**			
EGFR	cetuximab	7 (9.7)	-
VEGF	bevacizumab	7 (9.7)	-
**mAb + cytotoxic agents**			
EGFR + cytotoxic agents	cetuximab-oxaliplatin/5-FU/irinotecan	2 (2.8)	-
CD20 + cytotoxic agents	rituximab-vincristine-doxorubicine-cyclophosphamide	-	1 (6.3)
**mAb + multikinase inhibitor**			
VEGF + Raf, BRAF, VEGFR, PDGFR, c-KIT	bevacizumab–sorafenib	1 (1.4)	-
**Multikinase inhibitor**			
Bcr-Abl, c-KIT, PDGFR	imatinib mesylate	-	2 (12.5)
Raf, BRAF, VEGFR, PDGFR, c-KIT	sorafenib	4 (5.6)	-
VEGFR, PDGFR, c-KIT, FLT3, RET	sunitinib	10 (13.9)	-
VEGFR, PDGFR, c-KIT	axitinib	2 (2.8)	-
	pazopanib	2 (2.8)	-
**Serine–threonine kinase inhibitor**			
BRAF-MEK	dabrafenib–trametinib	1 (1.4)	-
**Tyrosine kinase inhibitor**			
EGFR-ALK	erlotinib–crizotinib	1 (1.4)	-
ALK	alectinib	1(1.4)	-
EGFR	erlotinib	2 (2.8)	-
	gefitinib	4 (5.6)	-
	osimertinib	1 (1.4)	-
Bcr-Abl	nilotinib	-	1 (6.3)
**Immunotherapy**			
PDL1	atezolizumab	1 (1.4)	
**Immunotherapy + multikinase inhibitor**			
PD1 + VEGFR, FGFR, PDGFR, c-KIT, RET	pembrolizumab-lenvatinib	1 (1.4)	

ALK: anaplastic lymphoma kinase; BRAF: V-Raf murine sarcoma viral oncogene homolog B1; c-KIT: tyrosine-protein kinase kit; EGFR: epidermal growth factor receptor; FLT3: Fms-like tyrosine kinase 3; Mab: monoclonal antibody; MEK: MAPK/ERK kinase; PDGFR: platelet-derived growth factor receptor; PD(L)1: programmed death (ligand) 1; RET: Ret proto-oncogene; VEGFR: vascular endothelial growth factor receptor; 5-FU: 5-fluorouracil.

**Table 3 cancers-14-01666-t003:** Clinical presentation and outcome of PI.

PI details—No (%)	Oncological (72 Cases)	Hematological (16 Cases)	Total (88 Cases)
Symptomatic	51 (70.8)	14 (87.5)	65 (73.9)
Asymptomatic	21 (29.2)	2 (12.5)	23 (26.1)
Perforation	10 (13.9)	1 (6.3)	11 (12.5)
Complications	6 (8.3)	6 (50.0)	12 (13.6)
Death	9 (12.5)	2 (12.5)	11 (12.5)
**PI site**—**no (%)**			
*Small intestine*	20 (27.8)	3 (18.8)	23 (26.1)
*Large intestine*	31 (43.1)	8 (50.0)	39 (44.3)
*Entire intestine*	17 (23.6)	2 (12.5)	19 (21.6)
*Unknown*	4 (5.6)	3 (18.8)	7 (8.0)
**GI surgery—no (%)**	5 (6.9)	1 (6.3)	6 (6.8)
**Therapy restarted—no (%)**	13 (18.1)	5 (31.3)	18 (20.5)
**Symptoms re-onset—no (%)**	5 (6.9)	1 (6.3)	6 (6.8)

GI: gastrointestinal; PI: pneumatosis intestinalis.

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
