# Peer review of "Pneumatosis Intestinalis Induced by Anticancer Treatment: A Systematic Review"

_cancers, 2022, doi:10.3390/cancers14071666_

Round 1

Reviewer 1 Report

This very interesting paper is a systematic review of every published case report or case series of pneumatosis intestinalis occuring in treated cancer patients.

Overall, this paper is very nicely written, appears succesful in delineating the clinical scenarios in which this very rare complication can occur, and carries some strong key messages. I also appreciate the very honest and clear description of the major limitations of the study.

I have some minor comments/questions :

regarding the abstract :

  • line 33 : "PI was mostly localized in respiratory system" : must be a mistake...
  • I would put the fact that 1/8 patients died of PI as a strong message. Consider something like "PI was fatal in 11 cases. In the remaining cases, symptoms can remain mild, or even be absent".
  • In my opinion, the message regarding risk of reoccurence at reintroduction of the drug is strong and should be present in the abstract.

The introduction is very clear. When speaking about the imaging techniques (lines 72-74), it would be very pedagogic to explain briefly the radiological findings to help clinicians recognize it. Clearly referencing or depicting a figure of PI would allow to completely understand this rare event in one article reading.

Line 94 : consider replacing "small observational studies" by "case reports or at best small case series".

line 204 : Perforation occurred ONLY in 11 cases. I would definitely remove "only". 1/8 risk of this severe complication is not insignificant. Just to compare, at a population level, countries screen every women every 2 years for breast cancer for a 12% lifetime risk... Besides, the same number of cases was fatal, and although these 11 were not the same, one can wonder if the fatal cases without perforation did not perforate later on.

Line 221 : consider adding "at diagnosis" after "even when asymptomatic".

Lines 278-282 : maybe too hazardous ? I would remove, and put this in the context of the further paragraph regarding comorbidities : did this patient treated with anti PD1 present a comorbidity ? I think a supplementary table resuming the important data per patient would be noteworthy, given these are all case reports. It is of course impossible to check for interactions, but this could give a sense regarding role of the drug (eg, in the case that was under mechanical ventilation).

Line 297 : few localized solid tumors (thus (neo)adjuvant setting) receive targeted therapies. This could explain why PI is more seen in metastatic setting. As this information should be present for all cases and it is a binary information, I would prudently explore here an interaction ?

I thus thank the authors, their manuscript was much informative but remained honest regarding what could be extracted from the quite poor original data. It should be also interesting for other readers.

Reviewer 2 Report

The aim of the study is stated to evaluate the causal factors associated with PI. This is not really what the team did or how this study was conducted. A different study design would be needed for this. Suggest revise the wording on line 105 to better represent the actual work presented.

Suggest that the inclusion/exclusion terms be clarified. Clinical trials and other such reports were excluded as well, is this correct?

Line 135: by oncology do you mean solid tumors?

Line 136: "stratification" technically is not really possible in this type of design. Do you mean that subgroups included tumor type, pharmacological therapy, and AE course? if these were separated, how were the latter two identified? Only tumor type is clearly stated as to how they were grouped.

Section 2.4. According to PRISMA guidelines, the screening and extraction should be done by two independent reviewers. Was this done? If so, please state how this was done. If not, state that this was a singular review and then revise the wording about being conducted in accordance with these guidelines.

I would like to see the full list of references of the included articles in the supplementary text, at a minimum. This is absolutely necessary to provide to the reader.

Line 183 suggests inference of causality and risk. This is not possible using this design. The fact a case report was published suggests there is something unique about the case that warranted this. It is incorrect to state "drugs most involved in PI occurrence." without a responsible denominator. This must be corrected here and elsewhere. Since there is no denominator, the comparative statements must be removed throughout the paper (e.g. Line 187, 229, 231 and many other places). This is extremely misleading and absolutely an incorrect approach to interpret these data. The limitation on line 323 is not at all sufficient when the authors have contradicted themselves via the presentation of data with regard to this stated limitation. 

Highly recommend that the limitation alluded to on line 323 be elaborated and explained to the reader. This is only descriptive and any statements suggesting more or higher or any sort of inferred association or comparative wording must be removed.

If level of risk or comparative work was intended, the team should have done a systematic review of clinical trial and/or observational data so that this could be evaluated with an appropriate denominator by which to make such comparisons. This is a major flaw of this work and precludes acceptance.

Round 2

Reviewer 2 Report

The authors have adequately addressed my comments with the revisions and the addition of the supplementary table. No further concerns.